# HPV Self-Sampling for Cervical Cancer Screening among Women Living with HIV in Low- and Middle-Income Countries: What Do We Know and What Can Be Done?

**DOI:** 10.3390/healthcare10071270

**Published:** 2022-07-08

**Authors:** Matthew Asare, Elakeche Abah, Dorcas Obiri-Yeboah, Lisa Lowenstein, Beth Lanning

**Affiliations:** 1Robbins College of Health and Human Services, Department of Public Health, Baylor University, Waco, TX 76798, USA; elakeche_abah1@baylor.edu (E.A.); beth_lanning@baylor.edu (B.L.); 2School of Medical Sciences, Department of Microbiology and Immunology, University of Cape Coast, Cape Coast, P.O. Box University Mail, Ghana; d.obiri-yeboah@uccsms.edu.gh; 3Department of Health Services Research, The University of Texas MD Anderson Cancer Center, Houston, TX 77030, USA; lmlowenstein@mdanderson.org

**Keywords:** HPV self-sampling, cervical cancer, women with HIV, low- and middle-income countries

## Abstract

Introduction. Self-sampling has the potential to increase cervical cancer (CC) screening among women with HIV in low- and middle-income countries (LMICs). However, our understanding of how HPV self-collection studies have been conducted in women with HIV is limited. The purpose of this scoping review was to examine the extent to which the HPV self-sampling has been applied among women with HIV in LMICs. Method: We conducted multiple searches in several databases for articles published between 2000 and January 2022. With the combination of keywords relating to HPV self-sampling, LMICs, and women with HIV, we retrieved over 9000 articles. We used pre-defined inclusion and exclusion criteria to select relevant studies for this review. Once a study met the inclusion criteria, we created a table to extract each study’s characteristics and classified them under common themes. We used a qualitative descriptive approach to summarize the scoping results. Results: A total of 12 articles were included in the final review. Overall, 3178 women were enrolled in those studies and 2105 (66%) of them were women with HIV. The self-sampling participation rate was 92.6%. The findings of our study show that 43% of the women with HIV in 8 of the studies reviewed tested positive for high-risk HPV (hr-HPV) genotypes, indicating 4 out of 10 women with HIV in the studies are at risk of cervical cancer. The prevalence of the hr-HPV in women with HIV was 18% higher than that of HIV-negative women. Most women in the study found the self-sampling experience acceptable, easy to use, convenient, and comfortable. Self-sampling performance in detecting hr-HPV genotypes is comparable to clinician-performed sampling. However, limited access (i.e., affordability, availability, transportation), limited knowledge about self-screening, doubts about the credibility of self-sampling results, and stigma remain barriers to the wide acceptance and implementation of self-sampling. In conclusion, the findings of this review highlight that (a) the prevalence of hr-HPV is higher among women with HIV than HIV-negative women, (b) self-sampling laboratory performance is similar to clinician-performed sampling, (c) the majority of the women participated in self-sampling, which could likely increase the cervical cancer screening uptake, and (d) women with HIV reported a positive experience with self-sampling. However, personal, environmental, and structural barriers challenge the application of self-sampling in LMICs, and these need to be addressed.

## 1. Introduction

Globally, persons with HIV, including those in low- and middle-income countries (LMICs), are living longer due to the wide availability of combination antiretroviral therapy (cART) [1,2,3,4]. In 2020, over 37.7 million people worldwide were living with HIV (including 1.5 million with new infections) [5,6]. The majority of the person with HIV live in low- and middle-income countries [6]. For instance, it was estimated that there were 20.6 million persons with HIV in East and Southern Africa regions in 2020, and in each, over 670,000 new HIV infections were reported [7]. Persons with HIV are at significantly high risk for developing Human papillomavirus (HPV)-related cancers, including cervical cancer (CC) [8]. Evidence showed that women with HIV have a six-fold higher risk of developing CC than their uninfected counterparts [9]. CC remains the number one cancer burden among women with HIV in low- and middle-income countries (LMICs) [10], with over 80% of the cancer burden concentrated in sub-Saharan Africans [9,11]. Additionally, women with HIV are at risk of developing CC up to 10 years earlier and require frequent screening [12,13].

Screening tests such as HPV tests, Pap tests, and visual inspection with acetic acid (VIA) are available for the early detection of CC risks. However, about 55 LMICs have no CC screening program [14]. Additionally, due to lack of coordination, the CC screening process in most LIMICs is sometimes considered an “opportunistic screening”, where a Pap test and VIA are requested for patients in clinics and hospitals either as part of general medical examination or for consultations related to or unrelated to CC [15,16]. The overall CC screening rate in LMICs is around 27%, which is very low [14,15,16]. The available screening participation rates in Ghana is 2.7% [17], in Kenya it is between 14% [18] and 17.5% [19], in Ukraine it is 30% [20], and in Nigeria it is 9.4% [21]. Implementation of CC screening programs in the LMICs has faced several complicated and context-specific challenges. The structural challenges include lack of funds, maldistribution of health workers, limited qualified personnel, and lack of infrastructure [15]. Individual-level barriers include cultural beliefs, perceived fear of screening procedures and adverse outcomes, societal stigmatization, embarrassment, lack of spousal support, lack of knowledge, cost of screening, privacy concerns, pain, misconceptions, lack of information, low prioritization of cancer screening, and the poor health status of women. HPV self-collection is a convenient way of testing that addresses many of the barriers women face, while also increasing screening participation, particularly in underscreened populations. The World Health Organization (WHO) recommends using HPV Deoxyribonucleic Acid (DNA) detection (including self-collection) as a primary cervical cancer screening test for women with HIV starting at the age of 25 years and subsequently every 3 to 5 years [22]. The WHO suggests using visual inspection with acetic acid (VIA) to triage women after positive HPV DNA test before treatment [22]. HPV self-collected cervicovaginal samples is a method where women self-collect vaginal samples and send them to the clinic or laboratory for analysis. Self-sampling has been promoted as an ideal option for low-resource areas because self-collection is more acceptable, relatively easy to implement, cost-effective, and sustainable in LMICs [23,24,25,26]. Offering women with HIV the option of Self-collected Cervicovaginal Samples (SCCS) at home could likely increase participation in CC screening programs [27]. Additionally, HPV self-collection is convenient, increases women’s sense of privacy, improves access in remote areas, decreases stigma and embarrassment, and reduces the potential financial (cost of self-sampling vs. the cost of clinician sampling) and logistical burden (i.e., cost of transportation and child care while attending clinician screening) for the patient [28]. Since the introduction of self-sampling methods, 11 LMICs have included self-sampling in their official programs [14].

In recent years, a few review articles including systematic reviews and meta-analyses have (a) compared the effectiveness of the self-collected sampling method with the effectiveness of the clinician-collected sampling method in the detection of high-risk HPV (hr-HPV) genotype [23,24,29,30,31], (b) evaluated acceptance and preference of self-sampling [31,32,33], and (c) assessed the knowledge of HPV and cervical cancer and acceptability of HPV self-sampling [34]. These previous review studies have contributed to our understanding that self-sampling is equally as effective as the clinician-collected sampling in detecting hr-HPV infections [23,24,29,30,31]. Based on those reviews, we also know that most women preferred self-sampling to clinician sampling, found self-collection acceptable [31,32,33], and many women have inadequate knowledge about self-sampling [34]. However, most of those review studies broadly focused on women in high-income countries and had a limited focus on women with HIV in LMICs. Understanding the extent to which self-sampling has been applied in women with HIV in LMICs is important for two reasons. First, due to poor health care infrastructure and inadequate qualified personnel, clinician-provided screenings such as HPV tests, Pap tests, and visual inspection with acetic acid (VIA) are few and far between in LMICs, making self-sampling a viable option to increase CC screening among women in LMICs [35,36,37]. Second, and most importantly, women with HIV bear a significant burden of CC and require regular screening [38,39], yet those women are underrepresented in standard CC screening [34,40]. It is, thus, critical to understand how this inexpensive, convenient, easy, and safe to use HPV self-sampling [41] has been implemented among this hard-to-reach population (i.e., women with HIV). A plethora of quantitative and qualitative studies have examined the effectiveness of self-sampling among women with HIV; however, to our knowledge, there are no reviews on the HPV self-collection behavior among women with HIV. The purpose of this scoping review was to examine the extent to which HPV self-sampling has been applied in addressing cervical cancer screening barriers among women with HIV in LMICs.

## 2. Materials and Methods

### 2.1. Search Strategy

We conducted multiple searches from 4 August 2021, to 31 January 2022, using MEDLINE, EMBASE, CINAHL, Google Scholar, Scopus, ERIC, Web of Science, and PsycINFO databases for published articles between 2000 and January 2022. Keywords used to identify articles included a combination of words relating to HPV self-sampling, HPV self-collection, HPV self-test, HPV self-administered sampling collection, women with HIV, low and middle income (LMICs), and cervical cancer screening. Where necessary we used the following filters to perform the search: (1) only published articles in peer-reviewed journals; (2) studies reported in English; and (3) studies that published the full text (if full text is not available during the literature search, we requested a copy of the full text through our institution interlibrary loan system).

### 2.2. Data Screening and Inclusion Criteria

A total of 9252 articles were retrieved using the search criteria. We used the PRISMA flowchart (Figure 1) to track the article screening process. First, we read the titles of the articles and excluded articles that were duplicated, were not peer-reviewed, focused on animal experiments or included animals in the study, were literature review papers, and/or focused on screening other than cervical cancer. The first step of the screening decreased the number of articles to 105. Second, we read the abstracts of the articles, and using the same selection process as in step one, this reduced the number of eligible articles to 30. Third, two research team members independently reviewed the full text of all 30 articles. An article was included in the final review if all the following criteria were met (1) studies conducted in LMICS; (2) studies that focused on self-screening or self-collection, and (3) studies that included women with HIV. Studies that enrolled 50% or more of women with HIV and used HIV-negative women as a comparison group were included in the final review. Both qualitative and quantitative studies were included in the final analysis. Studies that did not explicitly include or mention women with HIV and/or did not assess self-sampling as a study outcome were excluded from the final review.

### 2.3. Data Extraction

Once relevant studies that met the inclusion criteria were identified, we created a table to extract each study’s characteristics and we classified each study as follows: (a) author name, publication year, (b) study purpose, (c) study design, location, and recruitment method, (d) study sample size, demographic, and behavior, (e) theoretical framework, data collection self-sampling device, and self-sampling behavior performed, (f) study outcomes, and (g) study primary and secondary findings. Two reviewers extracted the data, and any differences of opinion were resolved through discussions. When agreement could not be reached, a third investigator was consulted.

### 2.4. Data Analysis

We used both quantitative and qualitative approaches to describe the study findings. We conducted quantitative descriptive analysis (i.e., frequency and proportion) using an Excel spreadsheet. A qualitative descriptive approach was used to summarize the results and categorize our scoping findings based on the context and commonalities across the reviewed studies [42,43]. This qualitative descriptive analysis is in keeping with the intent of scoping reviews that seek to identify the nature and extent of research evidence [44].

## 3. Results

### 3.1. Study Demographic Characteristics

Table 1 shows the findings from all the 12 articles which met the inclusion criteria and were included in the final review. Overall, 3178 women were enrolled in those studies and 2105 (66%) of the women were with HIV. Seven studies included women with HIV only, but five studies included both women with HIV and HIV-negative women, with the latter group used as a comparison group [36,45,46,47,48]. Study participants were women between the ages of 25 and 65 years and the sample size for the studies markedly varied. The sample size ranged from 21 to 1022. Many of the women included in the reviewed articles were attending hospitals/clinics for routine appointments.

### 3.2. Study Designs and Recruitment Methods

The majority of studies were conducted using quantitative methods (six were cross-sectional studies [47,48,49,50,51,52], three studies were quasi-experimental studies [41,45,53], and one was a prospective observational study [36]). Two were qualitative studies (interviews [46,54] and focus group discussion [46]), while one used mixed methods (i.e., focus group and survey [53]). Most of the cross-sectional studies involved short interventions or instructions where study participants received instructions about screening and/or about steps for using sampling kits [36,41,45,48,49,50,51,52]. However, the assessments of those outcomes were conducted at one point in time (snapshot assessments). The reviewed studies were implemented across eight LMICs, with most of the studies conducted in African countries, except for one that was conducted in Brazil [47]. Three studies were in Botswana [45,49,50], three in South Africa [36,41,51], and one each in Ghana [48], Côte d’Ivoire [54], Zimbabwe [52,53], and Uganda. Studies were generally similar in terms of study settings (clinics or hospitals) and recruitment methods (face-to-face). However, one study used mobile phone technology to recruit participants [54] and a few others did not report how they recruited study participants [36,46,52].

#### Self-Sampling Procedure

The reviewed studies’ participants performed self-sampling in most of the studies (9 out of 12 studies) [36,45,47,48,49,50,51,52,53]. In one of the studies, the participants received the intervention, followed by an examination of the self-sampling kits before completing the survey [41]. In two of the studies, the participants did not participate in self-sampling, nor did they see the self-sampling kits [46,54]. The most common sampling kits used in most of the studies were swabs, brushes, and tampons. Variations existed in the method used in collecting data for the study. While most of the studies included surveys, others included technology such as REDCap [49,50], electronic medical records [53], recorded interviews [46,47,53], and lab results [52]. While short intervention programs were implemented, none of the studies conducted pre- and post-intervention assessments. However, in one study, baseline assessments about participants’ knowledge and intention for self-screening were used, but no post-intervention assessment was conducted [53].

### 3.3. Theoretical Framework and Self-Sampling Approach

The majority of the studies were atheoretical. Three studies were developed using theoretical or conceptual frameworks to understand the HPV self-sampling application among women with HIV. The frameworks used by the researchers were the Health Belief Model (HBM) [54], Theory of Planned Behavior (TPB) [53], and social-ecological model (SEM) [46]. In the TPB study, a mixed-method (survey and interview) was used, while in HBM- and SEM-based studies, qualitative methods (i.e., focus group discussions and/or interviews) were used. The theory-based studies were formative studies, indicating the theories were used to understand women’s attitudes, perceptions, and beliefs about self-collection.

### 3.4. Outcome Variables

The most common outcomes of interest measured in the studies were knowledge, acceptability, and preference for HPV self-sampling, comparability between self-sampling and clinician-collected sampling in detecting hr-HPV genotypes, and the test for the prevalence of hr-HPV among the study participants. However, in two studies, researchers evaluated the facilitators and barriers to self-sampling, and in one study, mobile phone delivery of the test results to the women was evaluated as an outcome of interest.

The studies’ findings differed, with the most common themes reported being screening behavior, health outcome, the effectiveness of self-sampling methods in detecting HPV vs. clinician-collected sampling, facilitators and barriers, and women’s experiences with self-sampling.

#### 3.4.1. Screening Behavior

The screening behavior (defined as the proportion of study participants who completed the self-sampling) was assessed in nine studies [36,45,47,48,49,50,51,52,53]. The average screening rate across the nine studies was 92.6%. Eight of the studies [36,45,47,48,49,50,51,52] reported very high CC screening rates, ranging from 87% to 100%. However, one of the studies reported a screen rate of 51% [53].

#### 3.4.2. Health Outcomes

The health outcomes (defined as the prevalence of hr-HPV or HPV among the study participants) were determined in 8 out of the 12 reviewed articles [36,45,47,48,49,50,51,52,53]. The hr-HPV-positive prevalence among women with HIV was 43%, with Obiri-Yeboah et al. [48] reporting the minimum prevalence of 14% and Rodrigues et al. [47] reporting the maximum prevalence of 77.5%. Four of the studies [36,45,47,48] included compared the hr-HPV positivity rates among women with HIV with hr-HPV positivity rates among HIV-negative women and they found that the prevalence of the hr-HPV genotypes was higher among women with HIV (with prevalent rates between 14–77.5%) compared with HIV-negative women (with prevalent rates between 2–47%). The percentage score for the four studies showed hr-HPV prevalent at 37% (95%CI: −59.9–298.9) in women with HIV vs. hr-HPV prevalent at 19% (95%CI: −25.2–142.7) in HIV-negative women (about 18% higher in women with HIV). Four of the reviewed studies [49,50,51,53] without a comparison group reported that the prevalence of high-risk HPV among women with HIV ranged between 31% and 45%. Management of the positive results among the study participants was scarcely discussed.

#### 3.4.3. Self-Sampling vs. Clinician Sampling Comparison

The performance (positivity, sensitivity, and specificity) of self-sampling was evaluated against clinician sampling. Eight studies compared the positivity of self-sampling (defined as the ability to detect the presence of hr-HPV or HPV infection by the screening test), with clinician-performed sampling and found no significant difference between the two methods (self-sampling vs clinician). Only four out of the eight studies reported the positivity rates, and in those four studies [47,48,49,51], the overall HPV detection concordance ranged between 79.7% and 94.2%. A few of the reviewed studies evaluated the sensitivity (defined as the percentage of true-positive cases that are detected by the screening test) and specificity (defined as the percentage of true-negative cases that are negative by the screening test) of self-sampling and clinician-performed sampling. The results for the sensitivity and specificity were mixed. Two studies reported a strong sensitivity [Joseph et al. (82.1%) and Obiri-Yeboah et al. (92.6%)] and specificity [Joseph et al. (93.0%) and Obiri-Yeboah (93.0%)] agreement between the self-sampling and clinician sampling. However, Saidu et al. reported a high sensitivity rate (95.8%) for self-sampling but reported a low specificity rate (44.0%). Adamson et al.’s study found strong positivity agreement but they reported reduced sensitivity (77.4%) and specificity (77.8%) agreement between self-sampling and clinician sampling.

#### 3.4.4. Barriers and Facilitators

Three of the review studies [46,53,54] reported barriers and facilitators of self-sampling. The barriers are (a) personal barriers including lack of knowledge about the sample and the procedure for taking the sample, perceived competence about the ability to self-collect, fear of the consequences of self-collection results, being uncomfortable, and the financial burden [46,53,54], (b) environmental and/or cultural barriers emanating from stigma and discrimination [46,54], and (c) structural barriers, including access to care such as the cost of screening, transportation, lack of community-wide education, and insufficient resources for treatment or managing positive results [46,53]. Facilitators include higher knowledge about self-sampling, self-confidence, and fee removal [46].

#### 3.4.5. Women’s Experience

The acceptability and preference for self-sampling were assessed in most of the reviewed studies [47,48,50,51,53,54]. Overall, most women in the studies reported positive self-sampling experiences [47,48,50,51,53,54]. The acceptability of self-sampling among women was very high, with two studies [46,53] reporting that all the women indicated that self-sampling is an acceptable method and one other study [47] reporting that 87% of the women found self-sampling acceptable. Two studies [46,50] assessed women’s personal experience of taking self-sampling, and their most common responses were that self-sampling is easy to do, convenient, and comfortable. In three studies, the proportion of women who reported a preference for self-sampling to clinician sampling was 56.9%. Kohler et al. [50] reported the smallest percentage (19%) preference for self-sampling, Obiri-Yeboah et al. [48] reported a medium percentage (57.7%) preference, and Mahomed et al. [41] reported the largest percentage (94%) preference for self-sampling. Women’s preference for mobile phone delivery of the lab results was assessed in one study and 47% of participants preferred receiving results via mobile phone call [50].

## 4. Discussion

In this scoping review, we described the extent to which studies have applied self-sampling to increase CC screening among women with HIV in LMICs. Our main findings of the review can be summarized around the following themes: (a) screening behavior and health outcomes, (b) barriers to self-sampling, (c) procedures and methods used, and (d) theoretical framework.

### 4.1. Screening and Health Outcomes

#### 4.1.1. Screening Behaviors

The major finding is that many (8 out of 12) of the reviewed articles show high screening participation rates among the study participants, with those eight articles reporting a self-sampling screening rate of 92.6%. This finding is an indication that self-sampling will have the potential to increase CC screening and reduce CC death. This observation is consistent with the literature that found that self-sampling is associated with an increase in cervical cancer screening uptake [27]. Self-sampling makes community-level screening feasible, and thus, can help with the barrier of access to screening. Self-sampling can remove some of the cultural and personal barriers associated with clinician sampling and contributes to the increase in screening uptake [55].

#### 4.1.2. High-Risk HPV Prevalent

The findings of our study show that 43% of the women with HIV in 8 of the studies reviewed tested positive for hr-HPV genotypes, indicating 4 out of 10 WLWH in the studies are at risk of cervical cancer [36,45,47,48,49,50,51,52,53]. In a further analysis, one-third of the reviewed studies found that the prevalence of the hr-HPV genotypes among women with HIV was 18% higher than in HIV-negative women (women with HIV 37% vs. HIV-negative women 19%) [36,45,47,48]. These findings support the evidence that women with HIV are at higher risk of cervical cancer compare with HIV-negative counterparts [9].

#### 4.1.3. Self-Sampling Performance vs. Clinician Sampling

Another finding of our study is that self-sampling performance (i.e., positivity, sensitivity, and specificity) of detecting the presence of hr-HPV genotypes is comparable to clinician-performed sampling. Overall, the self-sampling performance in detecting the HPV-positive results showed a strong concordance with clinician sampling [47,48,49,51]. The concordance rates between the two sampling methods in four of the studies ranged between 79.7% and 94.2%, indicating that self-sampling is as effective as clinician sampling in detecting HPV infection [47,48,49,51]. Out of four studies that reported the sensitivity and specificity, only one study [36] reported a specificity rate of 44%, and the remaining three (75%) found that the sensitivity and specificity of self-sampling in detecting hr-HPV infection are similar to clinician sampling [48,51,52]. These findings agree with other reviews that reported that self-sampling is equally effective as the clinician-collected sampling in detecting hr-HPV infections [23,24,29,30,31]. The evidence for this is also seen in the fact that WHO has included self-sampling as an option in the 2021 guidelines [22].

#### 4.1.4. Women’s Experiences

The findings of our review show that most women had positive self-sampling experiences [46,47,48,50]. For instance, in four of the quantitative studies that assessed the women’s experiences with the HPV self-sampling, the majority of those women found the self-sampling experience acceptable [46,47,48,50]. The findings of the qualitative studies reviewed in our study offer further in-depth understanding as to why self-sampling is an acceptable option. Those qualitative studies revealed factors such as convenience, privacy, comfort, cost, and ease contribute to the popularity of self-sampling among women. Several review studies have come to a similar conclusion that most women find self-sampling easy and convenient to use [23,24,25,26,28].

Unlike a previous review study that found that women have a strong preference for self-sampling over clinician-performed sampling [32], our review findings showed mixed results. In a study by Kohler et al. [50], 19% of the women with HIV reported a preference for self-sampling, which is very low, but in the studies by Obiri-Yeboah et al. [48] and Mahomed et al. [41], 57.7% and 94% of the women, respectively, reported a preference for self-sampling over clinician sampling. Pierz et al.’s study included in this review elucidated plausible reasons for the mixed result regarding women’s preference for self-sampling. In that study, Pierz et al. explained that women’s perceived competence about their ability to self-collect own specimen was low. Again, Pierz et al. found that women with HIV were skeptic about the credibility of self-sampling results. These two factors could explain the reasons why women prefer clinician performed samples as opposed to self-sampling [46]. Another experience assessed by one of the reviewed studies is the use of mobile technology to deliver test results. Kohler et al. [50] found that 47% of women with HIV preferred receiving results via mobile phone call, and this shows the potential of mobile technology as a medium to promote self-screening.

#### 4.1.5. Barriers to Self-Sampling

Barriers to self-sampling were evaluated in some of the reviewed studies and the barriers identified were (a) personal barriers including lack of knowledge about the self-sample availability, its effectiveness, and the procedure for taking the sample, perceived competence about the participants’ ability to self-collect, fear of the consequences of self-collection results, uncomfortable feelings, financial burden, and doubts about the credibility of self-sampling results [46,53,54], (b) environmental and/or cultural barriers such as transportation stigma and discrimination emanating from friends and people within the community [46,54], and (c) structural barriers, including access to care such as the cost of screening, lack of community-wide education, and insufficient resources for treatment or managing positive results [46,53]. Facilitators for self-sampling include knowledge about self-sampling, self-confidence, and fee removal. The personal, environmental, and structural barriers identified lend credence to the common factors that have been identified to deter the general population from screening. Wong et al. identified similar factors as barriers to self-sampling among WLWH in high-income countries [34].

#### 4.1.6. Study Methods and Procedures

The other main findings worth discussing are the methods of recruitment, data collection, and study settings. Most of the studies used traditional methods (i.e., face-to-face contacts) of recruitment [41,45,48,49,50,51,54]. Face-to-face contact is effective in getting participants to studies [55], but this method is limited to reaching out to populations who happen to be at the recruitment sites at the time of recruitment. Mixed methods of data collection were used, including surveys and interviews and electronic media such as medical records and technology. The combination of these methods (survey and interviews) should continue to be used as they are effective methods. For the study setting, the majority of the studies recruited women from HIV hospitals and clinics. A few of the studies also applied theories to understand the screening behaviors of women.

#### 4.1.7. Limitations and Strengths

The limitation of this study is that we were able to analyze information that is published in peer-reviewed journals. There may be unpublished data that could be beneficial to this review, but because they are not published, we excluded them. In addition, only English language-based articles were reviewed, and since other languages are spoken in some of the LMICs, some data might have been missed. Another limitation is that the 12 publications identified were relatively small-scale studies, which impedes our ability to describe the extent to which self-sampling has been applied in addressing cervical cancer screening behavior. As more studies are published, future comprehensive literature reviews will be warranted to further elucidate the application of self-sampling in women with HIV. Despite the listed limitations, the study has several strengths. First, due to inadequate healthcare infrastructures and qualified personnel in LMICs [23,24], self-sampling is seen as a viable option to increase CC screening among women in LMICs [35,36,37]. However, studies that have applied self-sampling in women with HIV in LMICs have not been synthesized in the literature and this study seeks to close the gap in the literature. Second, and most importantly, women with HIV are disproportionately affected by cervical cancer and require regular screening [38,39], and yet, those women are underrepresented in standard CC screening [34,40]. Thus, highlighting how HPV self-sampling [41] has been implemented among this hard-to-reach population (i.e., women with HIV) is critical.

#### 4.1.8. Implications and Recommendations

The findings of the study show that self-sampling can increase screening participation and it is acceptable and efficacious. However, awareness of the availability and effectiveness of self-sampling is very low, and most women have low self-confidence in using self-sampling. Health practitioners and interventionists can implement behavioral interventions to create awareness at the individual and community levels. At the individual level, the interventions could help women build self-confidence about self-sampling. The interventions could emphasize the effectiveness of self-sampling results and emphasize that the purpose of early screening and detection is to detect the risk factor and not a diagnosis of cancer to allay the fears of women about the screening results leading to cancer diagnosis. In addition, it is important to emphasize that the increased awareness about self-sampling is aimed at decreasing the cervical cancer burden and preventable death.

At the community and population levels, interventions could help address the stigma-related barriers, create awareness about the burden of cervical cancer and about the effectiveness of HPV screening, and create a bottom-up advocacy group to demand policy changes regarding access (i.e., insurance coverage and availability of screening kits facilities) to screening and policies to address stigma. However, education at the individual and community levels alone is ineffective to bring about structural changes. Economic factors and varying healthcare priorities can limit the implementation of HPV self-sampling. Therefore, government, intergovernmental agencies, and non-governmental and philanthropic organizations can be mobilized to address the issue of screening accessibility and affordability barriers. At the governmental and policy levels, it is critical to emphasize that self-sampling is a highly cost-effective approach. Self-sampling overcomes the skilled personnel constraints faced by many LMICs, as only women who screen positive will require gynecological exams, and therefore, reducing the need for the specialized workforce [14]. Another emphasis is that many self-sampling devices do not require a cold chain and are stable after collection, minimizing the infrastructure and logistics required for transport to a central HPV testing facility [14].

While recruitments at the hospital facilities are the most common and effective practices to reach a selected few who are already motivated to take care of themselves, a comprehensive recruitment approach, including community outreach mobilization such as the use of mobile hospitals, churches, community centers, and involvement of opinion and community leaders, can be used to recruit participants. A practical approach that can increase self-sampling utilization at the community and population levels is the use of home or community visits by community health nurses [56]. Home visiting by community nurses is an integral component of healthcare systems in most LMICs to increase vaccination and other health behaviors [56,57,58] and the use of home-visiting strategies has been advocated by the WHO and UNICEF [57,59]. During the home or community visit, the community nurses can distribute the self-screening kits to the women in the community, explain how to collect the sample to the women, collect the sample back from the women, and take the samples to the lab for analysis. Though home visiting is labor-intensive, this approach can be critical to solving the transportation problem some women face and helping women to understand the self-collection procedure.

Another recruitment tool to increase self-screening uptake can be the use of mobile technologies and social media [60]. The use of mobile technology is associated with an increase in self-sampling and other sexual and reproductive health [60,61,62,63]. Mobile technology can be used as data collection tools, and for recruitment, delivering interventions, and delivering lab results. Social media and mobile technology, which have a far-reaching audience, and a larger population could be useful tools to reach women who would otherwise not patronize routine hospital visits. Another strategy to reach women beyond those in the hospitals is to use the mailed-in sampling kits approach. Mailed-in sampling kits have been applied in high-income countries and they are known to increase screening [64,65,66,67,68,69,70,71,72,73]. In LMICs, mailed sample kits coupled with mobile phone support could be a viable option. Transportation to medical centers for most women with HIV in LMICs is challenging, so mailing the kits to WLWH may not only alleviate the transportation burden, but will also offer the women the opportunity to self-collect the sample in the comfort of their home, provide privacy, and reduce stigma. While mailing the sample kits can be challenging in LMICs because of unreliable mailing systems, the proliferation of private couriers in LMICs [74,75] will make the mailing of the kits feasible. Some of the concerns about mailed-in kits are that some women in LMICs have low literacy and low self-confidence to self-collect samples. Thus, after the women received the mailed-in kits, phone calls, text messaging, and/or voicemail messaging systems can be used to support the women in taking the sample. Mobile technology interventions are ubiquitous even in the LMICs; more and more studies are using mobile technology [60,61,62,63]. Therefore, relying on mobile technology will help reach women who would otherwise be difficult to reach.

Additionally, an issue that was not sufficiently addressed is the management of women that receive positive results. Few studies mentioned how follow-up management for women was conducted [48]. It is important to ensure adequate follow-up and management of positive results from screening. Efforts should be made to encourage women to follow up with treatment. It will be counterproductive if women with HIV are encouraged to participate in cervical cancer screening, but women with positive screening results are not properly managed [14,76,77,78]. However, the cost of treatment may discourage most women from following up, as most of these women may not have insurance or cannot afford it. The problem of affordability calls for national policies in LMICS regarding screening and subsidies for treatment for women who cannot afford it.

Many of the studies were atheoretical and a few studies used theoretical or conceptual frameworks (HBM, TPB, and SEM) to understand the HPV self-sampling application among women with HIV. However, those theoretically based studies were limited in scope because the theories were used for feasibility studies alone. Previous studies suggest that theories and models are useful tools in recognizing and explaining the dynamics of behavioral change and in the development and implementation of intervention studies [79]. Theories and models help program planners to identify targets for behavioral change and methodologies to bring behavior change [79]. Furthermore, the application of theories and models enhances the replicability and scaling up of effective interventions [79]. Therefore, applying theoretical frameworks and logical constructs to understand behavioral patterns and guide effective interventions towards an increase in cervical cancer screening uptake among women with HIV in LMICs should be encouraged [80,81,82].

## 5. Conclusions

The findings of this review highlight that (a) articles that compared the prevalence of hr-HPV in women with HIV vs. women without HIV found a higher prevalence of hr-HPV in women with HIV than in women without HIV, (b) self-sampling performance (positivity, sensitivity, and specificity) in detecting hr-HPV genotypes is comparable to clinician-performed sampling, (c) the majority of the women participated in self-sampling, indicating that self-sampling has the potential to increase the cervical cancer screening uptake among women, and (d) women with HIV reported a positive experience with self-sampling and thus, found self-sampling acceptable, easy, and convenient to use. However, personal, environmental, and structural barriers challenge the application of self-sampling in LMICs. Recommendations are offered to increase self-sampling uptake in LMICs.

## Figures and Tables

**Figure 1 healthcare-10-01270-f001:**
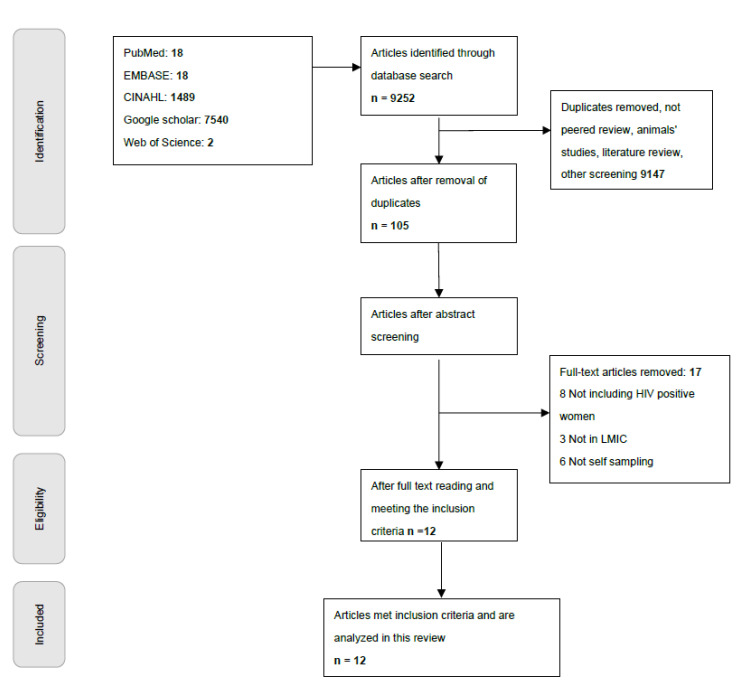
PRISMA flow chart of the search strategy.

**Table 1 healthcare-10-01270-t001:** Summary of self-sampling among women living with HIV in low- and middle-income countries.

Author/Year	Purpose	Sample Size Demographic Characteristics Behavior	Design Study Setting and Location Recruitment	Theory Data Collection Self-Sampling Device Performed	Outcome Variables	Primary/Secondary Findings
Saidu et al., 2021 [36]	To compare test performance of self- and clinician-collected samples in HIV-positive and HIV-negative women in South Africa	HIV-positive (n = 535) and HIV-negative (n = 586) women (Total 1121) Median age was 42 years Attending a primary health clinic and a teaching hospital	Prospective observational study with short instruction Hospitals in South Africa Recruitment method was not clearly described	No theory Documentation and Lab results Swab Conducted self-sampling	Self-sampling vs. clinician-collected samples in HIV-positive and HIV-negative women	HPV prevalence 25.1% for WLWH and 16.3% for HIV-negative women. There was good agreement (86.8%) between both methods of collection for detection of any hr-HPV. Sensitivity in WLWH 95.8% for self-sampling and 93.5% for clinicians. Lower specificity in SC samples for both HIV-positive (44.0%) and -negative women (77.5%).
Mahomed et al., 2014 [41]	To evaluate the acceptability of self-collection for cervical cancer screening	HIV-positive women (n = 106) The median age was 40 years Women attending clinics for care	Intervention with post-assessments but no control. Examined the device Hospital in South Africa. Face to face recruitment	No theory Survey Brush, lavager, and tampon. Not self-sampled	Self-collection device preference by women and willingness to use it for routine cervical cancer screening	In total, 94% of participants prefer self-sampling. Moreover, 75% of women from rural sites preferred cervical brush, while women from the urban clinic preferred the tampon-like plastic wand and lavage sampler.
Castle et al., 2020 [45]	To examine the feasibility of introducing HPV testing of self-collected vaginal samples and a hr-HPV screen-and-treat algorithm in Botswana	HIV-negative (n = 571) and HIV-positive (451) women (Total 1022) Median age WLWH (39 years) and negative (36 years) Women coming to the facilities for health care. No specific behavior description for women recruited in the community was given	Pilot Intervention study design with group education but no control group Health facilities in Botswana Research nurse contacted and community outreach events.	No theory Collected basic information but the method was not clearly stated Brush Self-sampled	hr-HPV prevalence among WLWH and HIV-negative women	Screening rate 99.7%. hr-HPV prevalence was 25.2% (95%CI = 21.2–29.4%) for HIV-negative women and 40.4% (95%CI = 36.3–44.5%) for WLWH. hr-HPV infection was common among all women in the study living in Botswana, to a greatest extent in WLWH than their HIV-negative counterparts.
Pierz et al., 2021 [46]	To assess and compare women’s perceptions and preferences for self- vs. provider-collected specimens	WLWH (n = 40) and HIV-negative (n = 40) women (Total 80) Women 25 years and above Attending the outpatient department for care.	Qualitative: Interviews and focus group Hospital in Cameroon Study nurses contacted participants, but the method of contact was not stated.	Socio-ecological model. Focus group discussions and interviews. Brush (Just for me) Not self-sampled	Perception of self-collection among WLWH and HIV-negative women; barriers and facilitators to obtaining and utilizing self-collected specimen	All participants indicated that self-sampling was an acceptable method of the specimen collection; barriers were lack of education about procedure and perceived competence about the ability to self-collect, fear and being uncomfortable, financial burden, stigma, pain and fear surrounding the provider sampling procedure, environmental context and stressors, and beliefs about consequences of self-collection.
Rodrigues et al., 2018 [47]	To evaluate the acceptability of cervicovaginal self-collection (CVSC) and prevalence of HPV in HIV-infected and HIV-uninfected women	HIV-infected (n = 41) and HIV-uninfected (n = 112) women (Total 153) Mean age was 36.9 years Underwent Pap smear	Cross-sectional study, but used a step-by-step explanatory pamphlet Health unit in Brazil Women were invited after the pap test, but no recruitment method was mentioned	No theory Interviews and lab results Brush Self-sampled	Self-sampling vs. clinician sampling. Acceptability of self-sampling and prevalence of HPV among HIV-infected and HIV-uninfected women	Overall acceptability of the self-sample was 87%. Prevalence of HPV and hr-HPV infection was 42.9% and 47.9% for HIV-uninfected and 97.6% and 77.5% for HIV-infected women, respectively. Positivity agreement 88.0% for HPV and 79.7% for hr-HPV. No sensitivity and specificity were assessed.
Obiri-Yeboah et al., 2017 [48]	To determine the acceptability, feasibility, and performance of alternative self-collected vaginal samples for HPV detection among Ghanaian women	WLWH (n = 97) and HIV-negative (n = 97) women (Total 194) Mean age was 44.1 years Women attending the HIV and outpatient clinics	Cross-sectional design with short instructions Hospital in Ghana Randomly recruited participants via face-to-face recruitment	No theory Survey and lab results Brush Conducted self-sampling	Self-sampling vs. clinician-collected (CC); preference sampling for women in specific socio-cultural settings	hr-HPV prevalence was 14.5%. Overall HPV detection concordance was 94.2%, similar between HIV-positive (93.8%) and HIV-negative women (94.7%). Highest sensitivity was among HIV-positive women and the highest specificity was among HIV-negative women. Sensitivity was 92.6% and specificity was 95.6%. Overall, 76.3% women found SC very easy/easy to obtain, 57.7% preferred SC to CC, and 61.9% felt SC would increase their likelihood to access cervical cancer screening
Elliott et al. 2019 [49]	Conducted the first assessment of self- versus provider-collected samples for hr-HPV testing using Xpert HPV in Botswana	Women living with HIV (n = 104) Median age 44 years age range 40–51 years, Attending routine appointments at the Hospital	Cross-sectional but intervention design with short instructions and no control group Hospital in Botswana Leaflets and face-to-face recruitment	No theory Survey and extraction of data from medical records. REDCap data collection Swab Self-sampled	hr-HPV positivity, any hr-HPV and type-specific HPV agreement between self and provider, and clinical outcomes among those testing positive for any hr-HPV	Screening rate was 99%. In total, 31 (30%) of 103 women tested positive for any hr-HPV. Overall agreement between self- and provider-collected samples for any hr-HPV was 92% with a κ of 0.80. In total, 10 of the 30 hr-HPV-positive women attending colposcopy had CIN 2+ (33%). No sensitivity and specificity tests were conducted.
Kohler et al. 2019 [50]	To assess the acceptability and preferences of HPV screening with self-sampling and mobile phone results delivery among women living with HIV (WLWH) in Botswana	Women living with HIV (n = 104) Median age 44 years age range 40–51 years, Attending routine appointments at the Hospital.	Cross-sectional but intervention design with short instructions and no control group Hospital in Botswana Leaflets and face-to-face recruitment	No theory Survey and extraction of data from medical records. REDCap data collection Swab Self-sampled	Knowledge, accessibility, and preferences of HPV self-sampling and mobile phone results delivery	Screening rate was 99%. Over 90% of participants agreed that self-sampling was easy and comfortable. In total, 95% were willing to self-sample again, but only 19% preferred self-sampling over a speculum exam for future screening. Moreover, 47% of participants preferred receiving results via mobile phone call. There were no positivity, sensitivity, and specificity tests.
Adamson et al., 2015 [51]	To access the acceptability and accuracy of cervical cancer screening using a self-collected tampon for HPV messenger-RNA testing among HIV-infected women	HIV-infected women (n = 325) Median age was 41.6 years Seeking care at a government HIV clinic	Cross-sectional study but intervention design with short instructions and no control group Hospital in South Africa Face-to-face recruitment	No theory Survey and medical record Tampon Self-sampled	Self-sampling vs. clinician sampling. hr-HPV prevalence, test positivity between two collection methods, accuracy and agreement of the two methods, acceptability of self-collection, and ease of use	Screening rate was 100%. Prevalence of 36.7% of hr-HPV. Positivity test (self-sampling 36.7% vs clinician 43.5%) was in agreement. Sensitivity was 77.4% and specificity was 77.8%. Tampon-based self-collection is acceptable to women and has similar hr-HPV mRNA positivity rates as clinician collection, but has reduced sensitivity and specificity compared to clinician collection
Joseph et al., 2021 [52]	To determine if self-collected samples could be used as an alternative to increasing coverage of cervical cancer screening programs	HIV-positive women (n = 280) Median age was 40 years. Attending pilot facilities for a routine appointment	Cross-sectional in nature after short instructions. Urban sites in Zimbabwe but not specifically identified. Specific recruitment method was not described	No theory Study staff collected data and entered them into online database and Lab results Swab Conducted self-sampling	Self-collected vs. clinician-collected samples	Results were found to have a good agreement: HPV prevalence was 43% for self-samples and 48% for clinician-collected samples. Sensitivity was 82.1% and specificity was 93.0%
Mitchell et al., 2017 [53]	To describe the knowledge and intentions of WHIV towards HPV self-collection for cervical cancer screening	HIV-positive women (n = 87) Age range was 30–60 years Attending the health unit for care	Intervention was conducted. A pre-intervention assessment was conducted Health unit in Uganda Phone calls were used to recruit participants and deliver results	Theory of planned behavior. Medical records, survey, and interview Swab Self-sampled	Knowledge and intentions towards HPV self-collection, factors related to HPV positivity	Screening rate was 51% (46% at the study clinic and 5% elsewhere). hr-HPV prevalence was 45%. In total, 98.9% did not think it necessary to be screened for cervical cancer. Almost all WHIV found self-collection to be acceptable; 40 women agreed to provide a sample at the HIV clinic. Drop-off kits are acceptable for the majority of the participants. Barriers include distance (travel was too far) and not having time to attend the screening
Mensah et al., 2020 [54]	To assess the preintervention acceptability of HPV screening among HIV-infected women in Abidjan, Côte d’Ivoire	HIV-positive (n = 21) Median age was 42 years, Attending a public clinic	Qualitative (Interviews) Public clinic in Abidjan, Côte d’Ivoire Recruitment method was face to face and over phone	Health belief model Recorded interviews No sample method No self-sampling conducted	Acceptability, knowledge, and beliefs about self-sampling	Barriers were the fear, stigma, poor knowledge of screening, and insufficient resources for treatment. Fees removal and higher levels of knowledge about cervical cancer and of the role of HIV status in cancer were found to facilitate screening. Self-confidence in self-sampling is low

## Data Availability

Not applicable.

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
