# Peer review of "HPV Self-Sampling for Cervical Cancer Screening among Women Living with HIV in Low- and Middle-Income Countries: What Do We Know and What Can Be Done?"

_healthcare, 2022, doi:10.3390/healthcare10071270_

Round 1

Reviewer 1 Report

Comments

The scoping review discuss HPV self-sampling for cervical screening specifically in HIV positive women in low- and middle-income countries. This interesting study is meaningful for the scholars and physicians specially having interest in public health, infectious diseases. Though the study is well constructed but some of the questions and minor revisions described below are needed.

Major Point

1.      Suggesting a concise title for the study: “HPV Self-Sampling for cervical screening among women living with HIV in low and middle-income countries. What do we know and what can be done?”

Abstract: Adequate

1.      P1L13 CCS abbreviation is only used rarely in the article, better keep it cervical cancer (CC) screening.

2.      P1L29 hr HPV place a hyphen in-between.

3.      P1L33 The author highlights that that the cervical cancer is a threat to every sexually active woman but for WLWH the threat increases. How does this review article cover the threat of HPV in terms of cervical cancer directly in here?

4.      P1L35 The author states that self-sampling is associated with increase in cervical cancer screening uptake but the review directly doesn’t cover the subject.

5.      P2L51 Remove coma before the reference.

6.      P2L56 Cited reference (#13) is on Hepatitis B (Replace in reference section).

7.      P2L63 Bring reference 15 at the end of sentence.

8.      P2L66 What is LIMCs? Isn’t it supposed to be LMICs?

9.      P2L76 CCS abbreviation is infrequently used better write it as CC screening.  

10.  P2L83 What is SCCS?

Materials and Methods

1.      P3L141 “also” within the sentence is grammatically incorrect.

2.      P4L164 Remove the second full stop.

3.      P4L180 What is LIMCs?

4.      P5L201 Theoretical framework after the heading is redundant.

Discussion

1.      P7L312 Hyphen is missing in hr HPV

2.      P9L389 The author states that self-sampling increases screening participation. Is this directly covered or was part of your review in here. Or it can?

3.      P9L423 What is LIMCs?

Conclusion

1.      P11L476 He? How does this review directly address the threat of cervical cancer increase by 18%? Isn’t it that the WLWH have increase infectivity rate of HPV as compared to the non-HIV in the studies reviewed?

2.      P11L478 hr HPV place hyphen in-between.

3.      P11L479 Do you mean “Self-sampling has the potential to increase the CC screening rate.”

References

1.      Follow the referencing style as recommended in instruction for the author (et al. should be used after 10 authors, write the journal name in abbreviated and italic form, and year in bold).

2.      P11L516 Ref 13 is on Hepatitis B.

3.      P11L516 Omit the duplicate serial number at the start of reference from Ref 13-40, 52, 59 and 60.

4.      P12L524 Add the page number of reference number 17.

5.      P12L560 Ref 33 and 34 are duplicate.

6.      P14L647 Use appropriate Ref for #76 and 77.

Table

1.      Caption the table

2.      Remove the serial number and reference the studies

Reviewer 2 Report

The authors conducted a thorough literature search to identify publications related to HPV self-sampling among women with HIV (WLWH) in low and middle income countries (LMICs). The authors did a nice job to introduce the disease burden of cervical cancer among WLWH in LMICs. The objective is to examine the extent to which HPV self-sampling has been applied in addressing cervical cancer screening barriers among WLWH in LMICs. If I am not miss-reading, the 12 publications identified by the authors were relatively small-scale studies to assess the agreement of self vs provider collected sample, acceptability and feasibility of self-sample or barrier of HPV self-sampling testing. These publications were from 8 different LMICs. The summary is helpful for the readers to know different studies had been done about HPV-self testing among WLWH in LMICs, however, I think population-based surveillance studies or additional evaluation studies might be needed to know in what extent that the HPV self-sampling is applied in those countries. For example, were the solutions applied to overcome the barriers identified? Was the self-sampling approach helped to increase the acceptance/uptake of cervical cancer screening among PLHIV?  Was the screening coverage increased in the country?

Reviewer 3 Report

The review emphasizes on the importance of self-sampling increases the participation of women living with HIV for preventive screening of high-risk cervical cancer. As the findings of the review showed 43% of WLWH patients test positive for high-risk cervical cancer genotypes, It also fills the knowledge in the HLWH community about the self-screening techniques in the detection of cervical cancer.

1. Findings of the review showed that 43% of WLWH patients test positive for high-risk cervical cancer genotypes- does it correlate with the clinical test results data? Is any patient clinical data available for this like the fatality rate among WLWH patients?

2. what is the age of patients included in the study? is there any correlation between the age and positivity rate?

3. does the data from the results of the study correlate with the high-income countries' data?

minor :

typos corrections needed.

Reviewer 4 Report

This paper tackles an important and neglected issue and I would very much like to see it published

Introduction

There is a good introduction setting out the issue's and the paper's significance.

Method

Methodologically, there seems to have been a careful stepwise selection process.

The paper mentions late that the language selectivity will have missed out some papers, but this is worth saying, and justifying, early on too. Perhaps the eg African francophone, Arabic, and Spanish/Portuguese literatures were thought to be relatively small? If so, say so.  

What constituted quasi-experimental and observational studies?

The types of studies are somewhat confusingly referred to later. I think it would be helpful if studies that did/not self-sample and that did/not compare techniques or women with/out HIV were 

People with HIV themselves and UNAIDS recommend the use of words and not initials in referring to people with HIV so 'women with HIV' should be preferred to WLWHIV throughout.

Findings

Outcomes, sensitivity, specificity: Where these were variable, what were possible differences within the studies (sample composition, country contexts, times within the HIV pandemic, nature of study) that may have played a part?

In general, findings variability needs to be related more to what may have generated it.Throughout findings it's also important to note which come from prospective studies.

Facilitation and barriers: This section is very short and non-specific about which papers showed what. I would suggest moving material into it from Discussion, which seems actually to contain Findings in this category.

Women's experience: This section is confusing because (see outcomes and my general point about findings) there's no account of study differences that might account for the variability.

Discussion

Some points eg screening rates, not clearly related to studies with actual/projected data. 

First part of 4.1.3 is quite hard to follow

Again it would help to disaggregate types of study in reporting conclusions being drawn from them

It would also be good if this section could more clearly indicate new conclusions emerging from this study, as well as points from prior studies that are being supported.

Sentence 2 of 4.1.8: This sentence is not warranted by what's just been argued - confusing.

Throughout this section, points need tying to prior studies. This only happens on the next page. In fact (see before) many points here seem rather to belong in Findings

The implications are very usefully addressed.

Conclusion

What is new about this study is not clear. I would also hope that the variability of findings could also be related at least provisionally to differences between the studies under consideration, so that differing results could have some kind of rationale.

Some English issues need to be addressed, and there are some mis-types. I haven't fully proofread - and that is needed - but small errors include:

1,2,2, used instead of 1,2,3 - lines 137-8

peer and not peered reviewed

prevalence, not prevalent.

One use of 'there' which should be 'their'

women's experience (plural)

Check plurals throughout (eg levels, interventions)

Also use possessives after author names when mentioning their papers (not done, in at least one place)

women WITH positive results

Round 2

Reviewer 2 Report

The authors did a nice job to introduce the disease burden of cervical cancer among women with HIV in low and middle income countries (LMICs). The objective of this paper is to examine the extent to which HPV self-sampling has been applied in addressing cervical cancer screening barriers among women with HIV in LMICs. Based on the authors’ summary table, the 12 publications in the 8 LMICs which the authors found were studies with relatively small sample sizes. Also, these studies were convenient samples and not population-based studies. My concern that the objective does not seem to be met remains. The authors need to be careful about generalizing the study findings from the samples to a larger papulation and taking average of the statistical estimates from different studies without taking into account the difference in sample sizes and variations between the studies.
